# Branched actin polymerization drives invasive protrusion formation to promote myoblast fusion during mouse skeletal muscle regeneration

Yue Lu[1]*, Tezin Walji[1], Pratima Pandey[1], Chuanli Zhou[2], Christa W Habela[3], Scott B Snapper[4], Rong Li[5,6], Elizabeth H Chen[1,7,8,9]*

[1]Department of Molecular Biology, University of Texas Southwestern Medical Center, Dallas, United States; [2]Department of Immunology, University of Texas Southwestern Medical Center, Dallas, United States; [3]Department of Neurology, Johns Hopkins University School of Medicine, Baltimore, United States; [4]Department of Pediatrics, Boston Children's Hospital, Boston, United States; [5]Department of Cell Biology, Johns Hopkins University School of Medicine, Baltimore, United States; [6]Mechanobiology Institute, National University of Singapore, Singapore, Singapore; [7]Department of Cell Biology, University of Texas Southwestern Medical Center, Dallas, United States; [8]Hamon Center for Regenerative Science and Medicine, University of Texas Southwestern Medical Center, Dallas, United States; [9]Harold C. Simmons Comprehensive Cancer Center, University of Texas Southwestern Medical Center, Dallas, United States

*For correspondence:
yue.lu@utsouthwestern.edu (YL);
elizabeth.chen@utsouthwestern.edu (EHC)

**Competing interest:** The authors declare that no competing interests exist.

## eLife Assessment

This study presents a **valuable** finding regarding the role of Arp2/3 and the actin nucleators N-WASP and WAVE complexes in myoblast fusion. The data presented is **convincing**, and the work will be of interest to biologists studying skeletal muscle stem cell biology in the context of skeletal muscle regeneration.

**Abstract** Skeletal muscle regeneration is a multistep process involving the activation, proliferation, differentiation, and fusion of muscle stem cells, known as satellite cells. Fusion of satellite cell-derived myoblasts (SCMs) is indispensable for generating the multinucleated, contractile myofibers during muscle repair. However, the molecular and cellular mechanisms underlying SCM fusion during muscle regeneration remain incompletely understood. Here, we reveal a critical role for branched actin polymerization in SCM fusion during mouse skeletal muscle regeneration. Using conditional knockouts of the Arp2/3 complex and its actin nucleation-promoting factors N-WASP and WAVE, we demonstrate that branched actin polymerization is specifically required for SCM fusion but dispensable for satellite cell proliferation, differentiation, and migration. We show that the N-WASP and WAVE complexes have partially redundant functions in regulating SCM fusion and that branched actin polymerization is essential for generating invasive protrusions at fusogenic synapses in SCMs. Together, our study identifies branched-actin regulators as key components of the myoblast fusion machinery and establishes invasive protrusion formation as a critical mechanism enabling myoblast fusion during skeletal muscle regeneration.

## Introduction

Skeletal muscle is a unique tissue composed of elongated multinucleated cells known as myofibers (*Frontera and Ochala, 2015*). In response to injury, skeletal muscle has the capacity to repair injured myofibers in a process called muscle regeneration. Muscle regeneration is dependent on the resident muscle stem cells, known as satellite cells (*Yin et al., 2013*; *Relaix et al., 2021*). Satellite cells are located between the myofiber plasma membrane and the basement membrane (BM), the latter of which is a layer of extracellular matrix material composed of collagen, glycoproteins, and proteoglycans (*Seale et al., 2000*; *Webster et al., 2016*). Satellite cells express high levels of Pax7, which is a paired domain-containing transcription factor, and remain quiescent under normal conditions (*Seale et al., 2000*; *Le Grand and Rudnicki, 2007*; *Cheung and Rando, 2013*; *Yin et al., 2013*). Upon injury, satellite cells are activated and then proliferate and differentiate into fusion-competent muscle cells to repair the injury (*Le Grand and Rudnicki, 2007*; *Cheung and Rando, 2013*; *Yin et al., 2013*; *Hindi and Millay, 2022*). Once the satellite cell-derived mononucleated muscle cells, which will be referred to as SCMs hereafter, fill the space within the BM remnants, known as ghost fibers, they would fuse with each other and/or with injured myofibers to regenerate the muscle (*Webster et al., 2016*; *Collins et al., 2024*). SCM fusion occurs rapidly between days 3.5 and 5 post injury (dpi) and persists till ~dpi 10 (*Collins et al., 2024*). Despite the importance of SCM fusion in skeletal muscle regeneration, the molecular and cellular mechanisms underlying SCM fusion during muscle regeneration remain poorly understood. To date, only two proteins, the bi-partite myoblast fusogens myomaker (MymK) (*Millay et al., 2013*) and myomixer (MymX)/myomerger/minion (*Bi et al., 2017*; *Quinn et al., 2017*; *Shi et al., 2017*; *Zhang et al., 2017*), have been shown to be required for SCM fusion in vivo (*Millay et al., 2014*; *Bi et al., 2018*). Identifying additional components of the SCM fusion machinery will not only facilitate our understanding of muscle regeneration but also provide more options to enhance muscle regeneration upon injury.

Studies in multiple organisms have provided significant insights into the evolutionarily conserved mechanisms underlying myoblast fusion during embryogenesis (*Chen, 2011*; *Kim et al., 2015a*; *Schejter, 2016*; *Deng et al., 2017*; *Kim and Chen, 2019*; *Lee and Chen, 2019*; *Petrany and Millay, 2019*). It has been demonstrated that embryonic myoblast fusion in *Drosophila*, zebrafish, and mouse embryos is mediated by an invasive podosome-like structure composed of actin-propelled membrane protrusions at the fusogenic synapse (*Sens et al., 2010*; *Luo et al., 2022*; *Lu et al., 2024*). The branched actin nucleator, the Arp2/3 complex (*Richardson et al., 2007*; *Berger et al., 2008*), and its actin nucleation-promoting factors (NPFs), the Neural Wiskott Aldrich Syndrome Protein (N-WASP [also known as Wasl]) (*Massarwa et al., 2007*; *Schäfer et al., 2007*; *Sens et al., 2010*; *Gruenbaum-Cohen et al., 2012*), and WASP-family verprolin-homologous protein (WAVE) (*Schröter et al., 2004*; *Richardson et al., 2007*; *Gildor et al., 2009*; *Sens et al., 2010*), are required for generating the invasive protrusions at the fusogenic synapse. Additional actin cytoskeletal regulators upstream of the NPFs also function in mammalian myoblast fusion during development, such as activators for N-WASP (Cdc42) and WAVE (Rac1) (*Vasyutina et al., 2009*), and the bi-partite guanine nucleotide exchange factor for Rac1 (Dock180 [also known as Dock1] and Elmo) (*Laurin et al., 2008*; *Tran et al., 2022*). A subunit of the WAVE complex (Nap1) has been shown to promote myoblast fusion in cultured C2C12 myoblasts (*Nowak et al., 2009*). Previous studies have shown actin-propelled protrusions between cultured SCMs (*Randrianarison-Huetz et al., 2018*), as well as membrane protrusions at the fusion sites of cultured SCMs (*Eigler et al., 2021*). Recent studies have revealed the mechanism underlying the formation of invasive protrusions at the fusogenic synapses – it takes the coordination of two Arp2/3 NPFs (WAVE and N-WASP) and two actin-bundling proteins (dynamin and WASP-interacting protein [WIP]) to generate mechanically stiff actin bundles that propel invasive protrusions (*Zhang et al., 2020a*; *Lu et al., 2024*). The essential function of the actin cytoskeleton in myoblast fusion has been further highlighted by the fact that each of the bi-partite muscle fusogens, MymK and MymX, requires a functional actin cytoskeleton to induce myoblast fusion (*Millay et al., 2013*; *Zhang et al., 2017*).

Despite all the previous studies, the potential function of branched actin polymerization in muscle regeneration in vivo has not been directly tested. Here, using satellite cell-specific knockout (KO) mice of Arp2/3 and NPFs, we show that branched actin polymerization is indispensable for muscle regeneration. In particular, Arp2/3 and NPFs are required for the formation of invasive protrusions during SCM fusion, but not satellite cell proliferation, differentiation, or migration. Thus, we have identified

new components of the SCM fusion machinery in vivo and demonstrated a critical role for branched actin-propelled invasive protrusions in skeletal muscle regeneration.

## Results

### SCMs populate the ghost fibers after macrophage departure at early stages of skeletal muscle regeneration

To examine SCMs after injury, we injured the tibialis anterior (TA) muscles by $BaCl_2$ injection (*Figure 1A*) and labeled the differentiating SCMs using an antibody against NCAM, a cell adhesion molecule highly expressed in these cells (*Capkovic et al., 2008*), and ghost fibers using an antibody against Laminin, a major component of the BM (*Webster et al., 2016*; *Figure 1B*). Since macrophages are present in the ghost fibers to clear the necrotic debris of the damaged myofibers (*Collins et al., 2024*), we also labeled macrophages with an antibody against MAC-2, a member of the lectin family expressed on the cell surface of macrophages (*Hohsfield et al., 2022*). At dpi 2.5, the differentiating SCMs and macrophages were two major cell populations residing within the ghost fibers, occupying 39.9% and 47.8% of the total volume, respectively. By dpi 3.5, SCMs filled 98.2% of the ghost fiber volume, whereas macrophages only accounted for 1.8% (*Figure 1C*), with most of the macrophages residing in the interstitial space outside of the ghost fibers (*Figure 1B*), which would account for the high overall number of macrophages in the regenerating muscle tissues in this time period (*Collins et al., 2024*). Of note, our confocal (*Figure 1—figure supplement 1A and B* and *Videos 1 and 2*) and transmission electron microscopy (TEM) (*Figure 1—figure supplement 1C*) analyses of regenerating TA muscles at dpi 3 also revealed narrow openings (~1 μm diameter) on the BM of the ghost fibers, through which macrophages (MAC-2⁺) with an ~20 μm diameter (as a round cell) were traversing, suggesting that macrophages enter and/or escape the ghost fibers by squeezing through tiny openings on the BM. By dpi 4.5, most of the SCMs have fused into multinucleated primary myofibers (*Figure 1B*; *Collins et al., 2024*).

### Branched actin polymerization is required for mammalian skeletal muscle regeneration

Given that the Arp2/3 complex-mediated branched actin polymerization is required for myoblast fusion during mouse embryogenesis (*Lu et al., 2024*), we asked whether the Arp2/3 complex is required for skeletal muscle regeneration in adults. Toward this end, we generated satellite cell-specific, tamoxifen-inducible KO mice for ArpC2, a subunit of the Arp2/3 complex (*Goley and Welch, 2006*), by breeding *Pax7*^CreERT2 mice (*Lepper et al., 2009*) with *Arpc2*^fl/fl mice (*Wang et al., 2016*). The conditional knockout (cKO) mouse line *Pax7*^CreERT2; *Arpc2*^fl/fl will be referred to as *Arpc2*-cKO hereafter. The littermates of the *Pax7*^CreERT2 mice without the floxed *Arpc2* allele were used as wild-type controls. To induce genetic deletion of *Arpc2* in satellite cells, we performed intraperitoneal injection of tamoxifen to the control and mutant mice every two days over a period of ten days (*Figure 2A*). The *Arpc2* KO in satellite cells was confirmed by western blot (*Figure 2—figure supplement 1*). Satellite cell-specific *Arpc2* KO did not affect TA muscle weight and size in uninjured muscle (*Figure 2—figure supplement 2*). However, muscle injury by $BaCl_2$ resulted in a significant decrease (87.7 ± 2.0%) in the cross-sectional area (CSA) of regenerated myofibers in *Arpc2*-cKO mice compared to their littermate controls at dpi 14 (*Figure 2B and C*) and dpi 28 (*Figure 2—figure supplement 3*). Consistent with this, the frequency distribution of CSA displayed a significant shift toward the small size in the mutant mice (*Figure 2D*). Taken together, these data demonstrate that the Arp2/3-mediated branched actin polymerization is essential for skeletal muscle regeneration.

### Branched actin polymerization is required for SCM fusion

To pinpoint the specific step of skeletal muscle regeneration – satellite cell proliferation, differentiation, migration, and SCM fusion – in which branched actin polymerization is required, we performed immunostaining using antibodies that specifically mark these steps. As shown in *Figure 3—figure supplement 1*, the percentages of muscle cells positive for the proliferation marker (Ki67) and the muscle differentiation marker (MyoG) in the injured TA muscles were similar between control and *Arpc2*-cKO mice (*Figure 3—figure supplement 1A–E*). In addition, live imaging analysis showed that cultured *Arpc2*-cKO SCMs exhibited normal migration and cell–cell contact behaviors (*Video 3*).

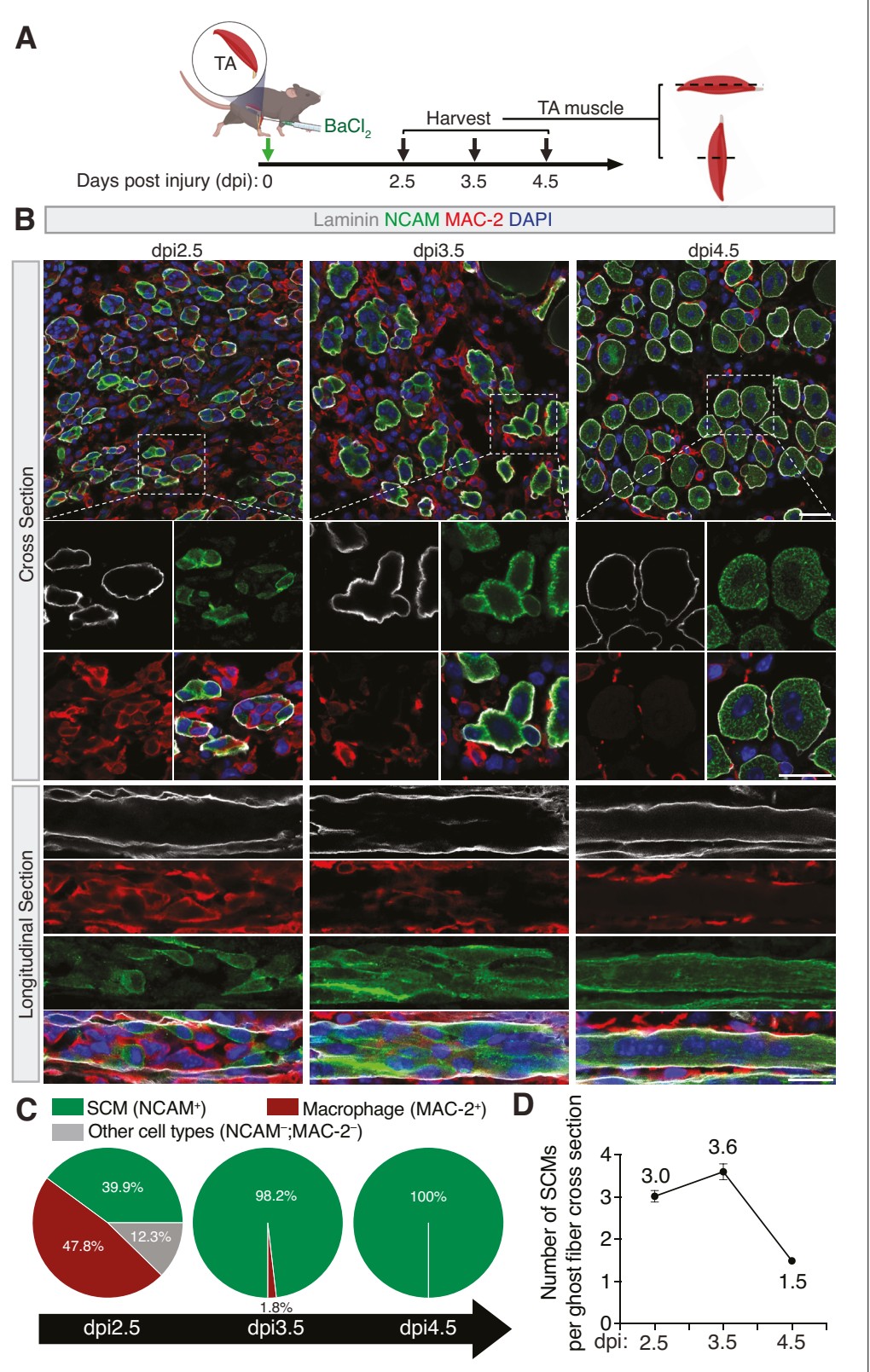

**Figure 1.** Spatiotemporal coordination of macrophages and satellite cell-derived myoblasts (SCMs) during skeletal muscle regeneration. (**A**) Diagram of the tibialis anterior (TA) muscle injury scheme. The TA muscles of the wild-type mice were injured by intramuscular injection of BaCl$_2$. The injured TA muscles were collected at dpi 2.5, 3.5, and 4.5 for cross and longitudinal sectioning and immunostaining. (**B**) Immunostaining with anti-Laminin,

*Figure 1 continued on next page*

*Figure 1 continued*

anti-NACM, and anti-MAC-2 of the cross and longitudinal sections of TA muscles at the indicated time points. Note the decrease in the macrophage number within the ghost fiber at dpi 3.5 (compared to dpi 2.5), and the fusion of SCMs between dpi 3.5 and 4.5. Scale bars: 20 µm. (**C**) Quantification of the percentage of macrophages and differentiated SCMs within ghost fibers at the indicated time points. n=3 mice were analyzed for each time point and >98 ghost fibers in each mouse were examined. Mean ± s.d. values are shown. (**D**) Quantification of the number of differentiated SCMs in a single cross-section of a ghost fiber at indicated time points. n=3 mice were analyzed for each time point and >98 ghost fibers in each mouse were examined. Mean ± s.e.m values are shown.

The online version of this article includes the following figure supplement(s) for figure 1:

**Figure supplement 1.** Macrophages extravasate the ghost fibers by traversing the basement membrane (BM).

These results demonstrate that branched actin polymerization is dispensable for satellite cell proliferation, differentiation, and migration during skeletal muscle regeneration. Thus, the reduced muscle size in the *Arpc2* mutant mice is likely due to defects in SCM fusion.

To monitor the SCM fusion phenotypes, we examined the regenerating TA muscles of the control and *Arpc2*-cKO mice at dpi 4.5, when myoblast fusion leading to primary myofiber formation is mostly completed (*Figure 1B*; *Collins et al., 2024*). While most of the SCMs within the ghost fibers had fused in the control animals, the ghost fibers in the *Arpc2*-cKO mice contained differentiated (NCAM⁺), but mostly unfused, SCMs, which were readily observed in cross-sections, comparable to those in the *Mymx*-cKO mice (*Figure 3A–C*, *Figure 3—figure supplement 2A*). Consistent with this, the frequency distribution of SCM numbers in a cross-section per ghost fiber in the *Arpc2*-cKO and *Mymx*-cKO mice displayed a dramatic shift toward higher numbers compared with that of the wild-type mice (*Figure 3D*). These results indicate that the actin cytoskeleton plays an essential role in SCM fusion as the fusogenic proteins. Interestingly, expression levels of the fusogenic proteins, MymK and MymX, in the TA muscle of *Arpc2*-cKO mutant mice were either similar to or higher compared to that of wild-type mice (*Figure 3—figure supplement 1F–H*), suggesting that the fusion defect in the *Arpc2*-cKO mutant mice was not due to a lack of fusogen expression. Consistent with this, cultured *Arpc2*-cKO SCMs exhibited a severe fusion defect despite undergoing normal differentiation (*Figure 3E–G*) and pharmacologically inhibiting Arp2/3 with CK666 also led to a similar fusion defect (*Figure 3—figure supplement 2B–D*). In addition, cell-mixing experiments using wild-type and *Arpc2*-cKO SCMs showed that *Arpc2*-cKO SCMs failed to fuse with wild-type cells (*Figure 3H and I*), indicating

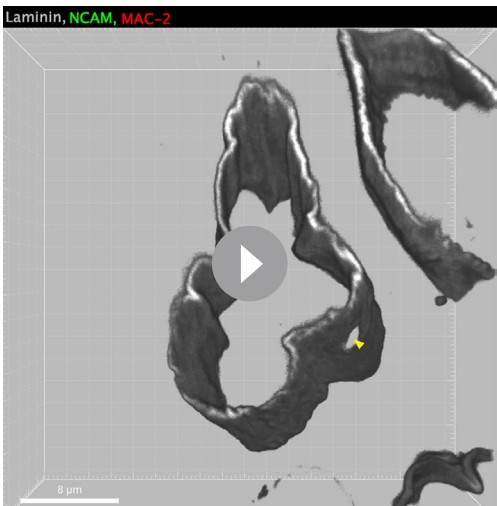

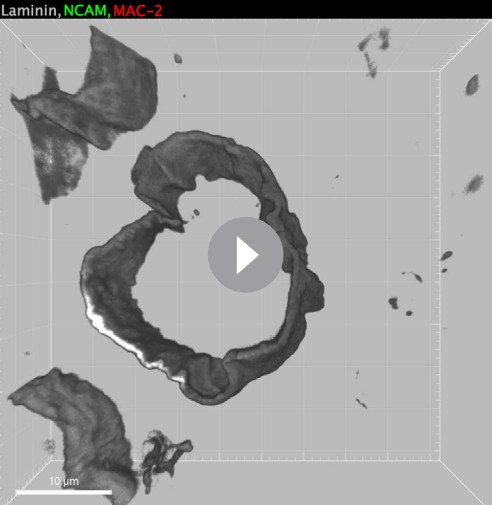

**Video 1.** Macrophages extravasate the ghost fibers by traversing the BM–ghost fiber 1. Representative 3D reconstruction of confocal z-stacks of TA muscle at dpi 3.5. The small opening on the BM is indicated by yellow arrowheads, and the transversing macrophage is indicated by magenta arrows.

https://elifesciences.org/articles/103550/figures#video1

**Video 2.** Macrophages extravasate the ghost fibers by traversing the BM–ghost fiber 2. Representative 3D reconstruction of confocal z-stacks of TA muscle at dpi 3.5. The small opening on the BM is indicated by yellow arrowheads, and the transversing macrophage is indicated by magenta arrows.

https://elifesciences.org/articles/103550/figures#video2

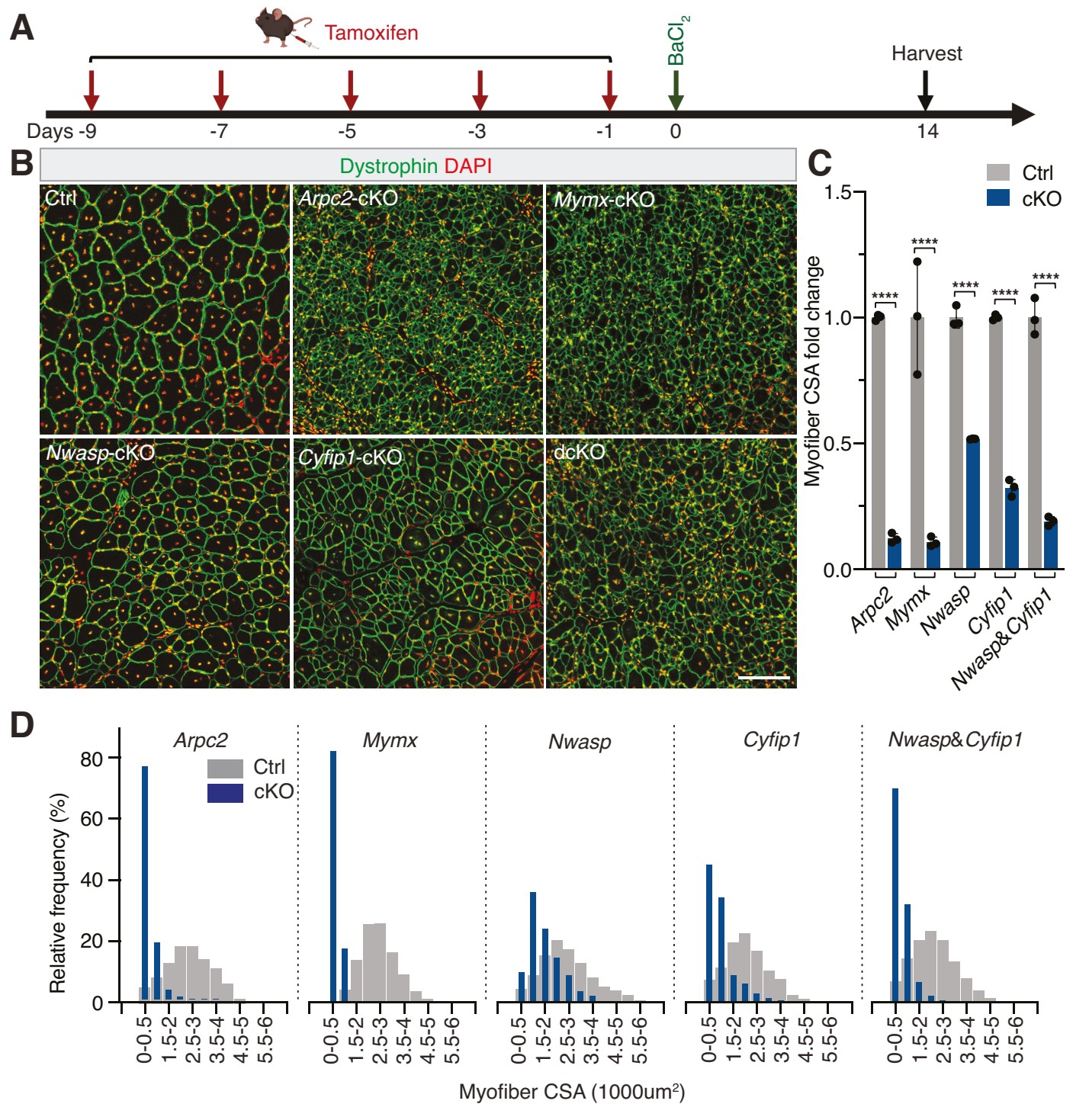

**Figure 2.** Branched actin polymerization is required for skeletal muscle regeneration. (**A**) Schematic diagram of tamoxifen and BaCl$_2$ treatment and subsequent cross-sectional area (CSA) analysis at dpi 14. (**B**) Dystrophin and DAPI staining of the cross-sections of TA muscles at dpi 14 from the control (Ctrl) and mutant mice. Note that the myofiber CSA is moderately decreased in *Nwasp*-cKO and *Cyfip1*-cKO mice and severely reduced in dcKO, *Arpc2*-cKO, and *Mymx*-cKO mice. Scale bar: 100 μm. (**C**) The fold change of myofiber CSA in mutant mice *vs.* control mice. n=3 mice were analyzed for each time point and >200 fibers in each mouse were examined. Mean ± s.d. values are shown in the bar graph, and significance was determined by two-tailed Student's *t*-test. ****p<0.0001. (**D**) Frequency distribution of myofiber CSA of TA muscles in the control and mutant mice at dpi 14. n=3 mice of each genotype were examined and >200 ghost fibers in each mouse were examined.

The online version of this article includes the following source data and figure supplement(s) for figure 2:

*Figure 2 continued on next page*

that branched actin polymerization is required in both fusing partners. Taken together, these results demonstrate that branched actin polymerization is required in SCMs for their fusion during skeletal muscle regeneration. The severe myoblast fusion defects observed in early stages of regeneration (e.g., dpi 4.5) provide a good explanation for the presence of thin muscle fibers in *Arpc2*-cKO mice at dpi 14 (*Figure 2B and C*) and dpi 28 (*Figure 2—figure supplement 3*). These thin muscle fibers could be either elongated mononucleated muscle cells or multinucleated myofibers each containing a small number of nuclei due to occasional fusion events (comparable to those in *Mymx*-cKO muscles) (*Figure 2B and C*; *Figure 2—figure supplement 3*). Whether Arp2/3 and branched actin polymerization may play a role in the growth and/or maintenance of post-fusion multinucleated myofibers requires future loss-of-function studies to inactivate *Arpc2* using a myofiber-specific Cre driver.

## N-WASP and WAVE families have partially redundant functions in regulating SCM fusion

Activation of the Arp2/3 complex requires the actin NPFs, including the WASP and WAVE family of proteins (*Goley and Welch, 2006*). To examine their potential functions in mammalian muscle regeneration, we generated single and double cKO mice for N-WASP [the WASP family member with high expression in SCMs *Lu et al., 2024*] and CYFIP1 [a subunit of the WAVE complex *Eden et al., 2002*], respectively. The cKO mouse line $Pax7^{CreERT2}$; $Nwasp^{fl/fl}$ will be referred to as *Nwasp*-cKO, $Pax7^{CreERT2}$; $Cyfip1^{fl/fl}$ as *Cyfip1*-cKO, and $Pax7^{CreERT2}$; $Nwasp^{fl/fl}$; $Cyfip1^{fl/fl}$ as dcKO hereafter. Target protein knockouts in SCMs were confirmed by western blot (*Figure 2—figure supplement 1*).

For the single cKO mice, immunostaining revealed a moderate but significant reduction of TA myofiber CSA at dpi 14 by 48.2 ± 0.1% in *Nwasp*-cKO and 67.7 ± 3.3% in *Cyfip1*-cKO mice, respectively, compared to their littermate controls (*Figure 2B and C*). The myofiber CSA of dcKO mice further decreased to 80.9 ± 1.8%, comparable to the 87.7 ± 2.0% observed in the *Arpc2*-cKO mice (in which both N-WASP and WAVE complexes are defective) and to the 89.3 ± 1.9% in the *Mymx*-cKO mice (*Figure 2B and C*). Moreover, the *Nwasp* and *Cyfip1* single KO mice exhibited moderate myoblast fusion defects at dpi 4.5, which were exacerbated in dcKO mice (*Figure 3B–D*), despite normal satellite cell proliferation and differentiation, as well as the persistent fusogenic protein expression in the dcKO mice (*Figure 3—figure supplement 1A–H*). Thus, our data revealed partially redundant functions between N-WASP and WAVE NPFs in promoting myoblast fusion during skeletal muscle regeneration.

## Branched actin polymerization promotes invasive protrusion formation during SCM fusion

To investigate the mechanism by which branched actin polymerization regulates SCM fusion during muscle regeneration, we first examined the

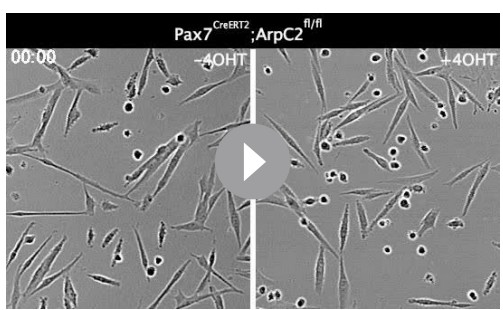

**Video 3.** Branched actin polymerization is not required for SCM migration during differentiation. Time-lapse imaging of control and *Arpc2*-cKO SCMs at 24 hours in DM. The SCMs isolated from *Arpc2*-cKO mice were maintained in GM without or with 2 μM 4OH-tamoxifen (4OHT) for 10 days. Subsequently, the cells were plated in 70% confluence in GM. After 24 hours, the cells were cultured in DM for 12 hours, followed by live cell imaging. Note that the *Arpc2* KO SCMs were able to migrate normally, although their fusion was significantly impaired. The time interval is 5 minutes.
https://elifesciences.org/articles/103550/figures#video3

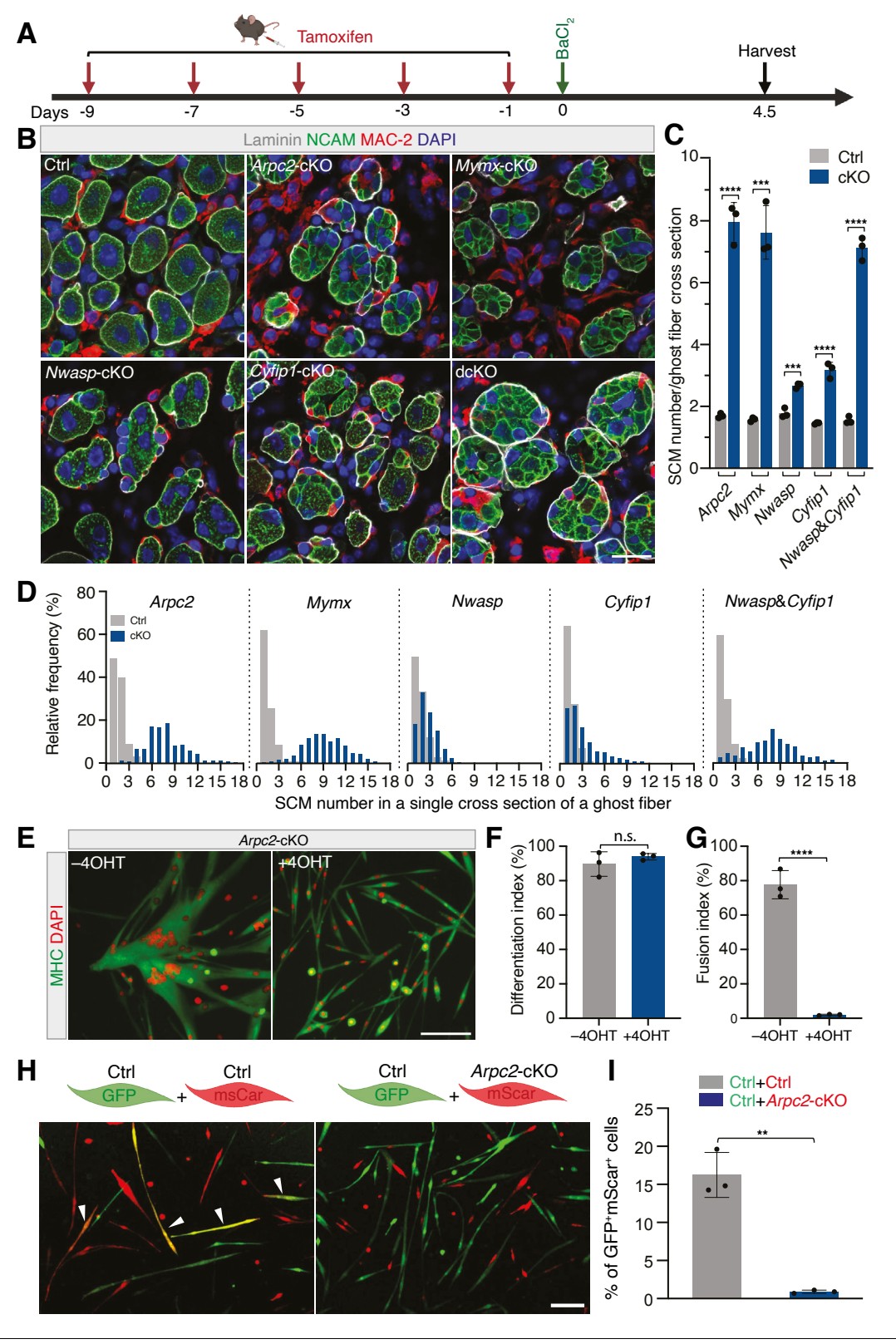

**Figure 3.** Branched actin polymerization is required for satellite cell-derived myoblast (SCM) fusion. (**A**) Schematic diagram of tamoxifen and BaCl₂ treatment and subsequent SCM number analysis at dpi 4.5. (**B**) Immunostaining with anti-laminin, anti-NCAM, and anti-MAC-2 of the cross-sections of TA muscles at dpi 4.5 from the Ctrl and mutant mice. Note that each ghost fiber in the Ctrl mice contained 1–2 centrally nucleated myofibers at dpi 4.5,

*Figure 3 continued on next page*

*Figure 3 continued*

indicating the near completion of SCM fusion. The ghost fibers in *Nwasp*-cKO and *Cyfip1*-cKO mice contained more SCMs, indicating impaired SCM fusion. Note that even more SCMs were seen in dcKO, *Arpc2*-cKO, and *Mymx*-cKO mice. Scale bar: 20 µm. (**C**) Quantification of the SCM number in a single cross-section of a ghost fiber from TA muscles of the Ctrl and mutant mice at dpi 4.5. n=3 mice were analyzed for each time point and >80 ghost fibers in each mouse were examined. Mean ± s.d. values are shown in the bar graph, and significance was determined by two-tailed Student's *t*-test. ***p<0.001 and ****p<0.0001. (**D**) Frequency distribution of SCM number in a single cross-section of a ghost fiber from TA muscles of the mutant mice and their littermate Ctrl. n=3 mice of each genotype were analyzed and >80 ghost fibers in each mouse were examined. (**E**) ArpC2 is required for SCM fusion in cultured cells. The satellite cells isolated from *Arpc2*-cKO mice were maintained in GM without or with 2 µM 4OH-tamoxifen (4OHT) for 10 days. Subsequently, the cells were plated at 70% confluence in GM. After 24 hours, the cells were cultured in DM for 48 hours, followed by immunostaining with anti-MHC and DAPI. Note the robust fusion of the control (–4OHT) SCMs and the severe fusion defects in *Arpc2*-cKO (+4OHT) SCMs. Scale bar: 100 µm. (**F, G**) Quantification of the differentiation index (% of nuclei in MHC$^+$ cells vs. total nuclei) and fusion index (% of nuclei in MHC$^+$ myotubes with ≥3 nuclei vs. total nuclei) of the two types of cells shown in (**E**). n = 3 independent experiments were performed. Mean ± s.d. values are shown in the bar graphs, and significance was determined by two-tailed Student's *t*-test. ****p<0.0001; n.s: not significant. (**H**) ArpC2 is required in both fusion partners. Fluorescence images from cell-mixing experiments using differentially labeled SCMs are shown. The satellite cells isolated from *Arpc2*-cKO mice were infected with retroviruses encoding GFP or mScarleti (mScar). Next, the GFP$^+$ cells were maintained in GM for 10 days (Ctrl GFP$^+$ cells), and the mScar$^+$ cells were maintained in GM without (Ctrl mScar$^+$ cells) or with 2 µM 4OH-tamoxifen (*Arpc2*-cKO mScar$^+$ cells) for 10 days. Subsequently, the Ctrl GFP$^+$ cells were mixed with Ctrl mScar$^+$ cells or with *Arpc2*-cKO mScar$^+$ cells with a ratio of 1:1 and plated at 70% confluence in GM. After 24 hours, the cells were cultured in DM for 48 hours followed by direct fluorescent imaging. Arrowheads indicate syncytia derived from both GFP and mScar cells. Scale bar: 100 µm. (**I**) Percentage of GFP$^+$mScar$^+$ syncytia in total cells shown in (**H**). n=3 independent experiments were performed. Mean ± s.d. values are shown in the bar graph, and significance was determined by two-tailed Student's *t*-test. **p<0.01.

The online version of this article includes the following source data and figure supplement(s) for figure 3:

**Figure supplement 1.** Branched actin polymerization is not required for satellite cell proliferation, differentiation, or fusogenic protein expression.

**Figure supplement 1—source data 1.** PDF file containing original western blots for *Figure 3—figure supplement 1F*, indicating the relevant bands and treatments.

**Figure supplement 1—source data 2.** Original files for western blot analysis displayed in *Figure 3—figure supplement 1F*.

**Figure supplement 2.** Branched actin polymerization is required for satellite cell-derived myoblast (SCM) fusion.

cellular structure at the fusogenic synapse of cultured SCMs. Live cell imaging of cultured SCMs expressing Arp2-mNeongreen (mNG) and LifeAct-mScarleti (mScar) at day 2 in differentiation medium (DM) revealed Arp2- and F-actin-enriched finger-like protrusions projecting from the invading cells into their fusion partners (receiving cells) at the fusogenic synapse prior to cell membrane fusion (*Figure 4A and A'* and *Video 4*). Consistent with this, TEM analysis of the TA muscle at dpi 3.5 in wild-type mice revealed finger-like protrusions projected by SCMs invading their neighboring cells (20.3 ± 16.5% of SCMs exhibited invasive protrusions, n=83 SCMs from 20 ghost fibers examined) (*Figure 4B and C*). The average length and width of the invasive finger-like protrusions were 422±200 nm and 121±73 nm, respectively (*Figure 4—figure supplement 1*, n=32 invasive protrusions examined). In contrast, muscle cells in *Arpc2*-cKO mice seldom projected invasive protrusions (*Figure 4B and C*, 0.5 ± 2.2% of SCMs exhibited invasive protrusions, n=147 SCMs from 20 ghost fibers examined), whereas protrusions in SCMs of *Mymx*-cKO mice appeared normal (*Figure 4B and C*, 24.1 ± 15.6% of SCMs exhibited invasive protrusions, n=93 SCMs from 20 ghost fibers examined; *Figure 4—figure supplement 1*, n=29 invasive protrusions examined). Therefore, branched actin polymerization, but not the fusogenic protein MymX, is required for invasive protrusion formation to promote myoblast fusion during adult muscle regeneration.

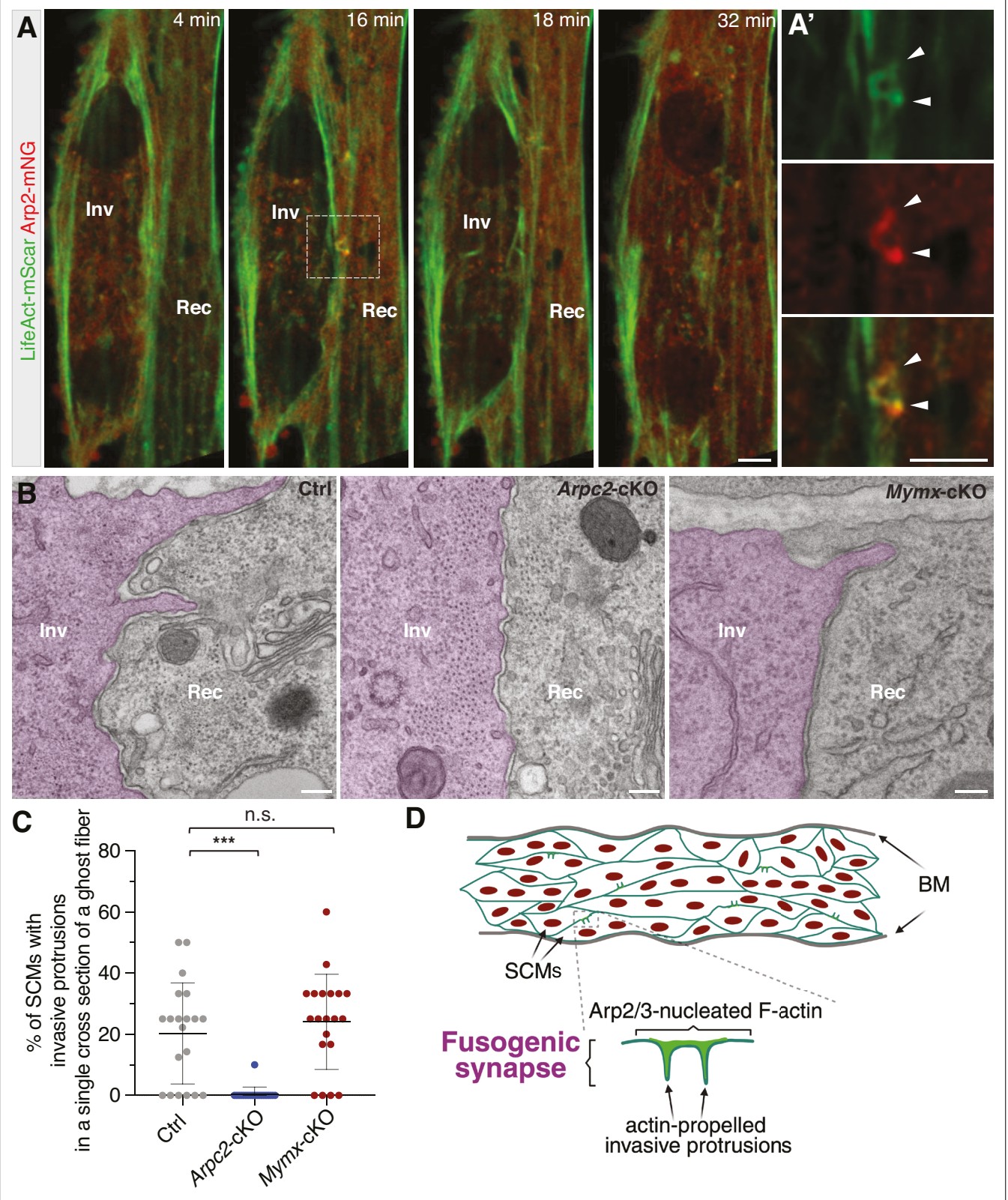

**Figure 4.** Branched actin polymerization is required for invasive protrusion formation during satellite cell-derived myoblast (SCM) fusion. (**A**) Still images of a fusion event between two LifeAct-mScar and Arp2-mNG co-expressing SCMs (see *Video 4*). The boxed area is enlarged in (**A'**). Note the presence of two invasive protrusions (16 minutes, arrowheads) enriched with LifeAct-mScar and Arp2-mNG at the fusogenic synapse. n=8 fusion events were observed with similar results. Scale bar: 5 µm. (**B**) TEM of TA muscle cells in wild-type Ctrl, *Arpc2*-cKO, and *Mymx*-cKO mice at dpi 3.5. The invading

*Figure 4 continued on next page*

*Figure 4 continued*

SCMs are pseudo-colored in light magenta. Note the finger-like protrusions projected by SCMs invading their neighboring cells in Ctrl and *Mymx*-cKO, but not in the *Arpc2*-cKO, mice. Scale bars: 500 nm. (**C**) Quantification of the percentage of SCMs with invasive protrusions in a single cross-section of a ghost fiber in the mice with genotypes shown in (**B**) at dpi 3.5. At least 83 SCMs from n=20 ghost fibers in each genotype were quantified. Mean ± s.d. values are shown in the dot plots, and significance was determined by two-tailed Student's *t*-test. ***p<0.001; n.s.: not significant. (**D**) A model depicting the function of Arp2/3-mediated branched actin polymerization in promoting invasive protrusion formation to promote SCM fusion during skeletal muscle regeneration. BM: basement membrane.

The online version of this article includes the following figure supplement(s) for figure 4:

**Figure supplement 1.** Fusogenic protein MymX is not required for invasive protrusion formation.

## Discussion

In this study, we show that the Arp2/3 complex-mediated branched actin polymerization is indispensable for SCM fusion, but not for satellite cell proliferation, migration, or differentiation during muscle regeneration. The Arp2/3 NPFs, N-WASP and WAVE, exhibit partially redundant functions in regulating SCM fusion. Our live cell imaging and electron microscopy analysis revealed actin-propelled invasive protrusions at the fusogenic synapses of SCMs, and our genetic analysis demonstrated a requirement for branched actin polymerization in generating these protrusions. Taken together, we propose that branched actin polymerization promotes mammalian muscle regeneration by facilitating the formation of invasive protrusions at the fusogenic synapse (*Figure 4D*).

Studies in multiple organisms, including *Drosophila*, zebrafish, and mouse, have demonstrated that myoblast fusion during embryogenesis is mediated by actin-propelled invasive membrane protrusions (*Sens et al., 2010*; *Jin et al., 2011*; *Duan et al., 2012*; *Duan et al., 2018*; *Luo et al., 2022*; *Lu et al., 2024*). These protrusions enhance the plasma membrane contact areas between the fusion partners and increase the mechanical tension of the fusogenic synapse to promote fusion (*Chen, 2011*; *Shilagardi et al., 2013*; *Kim et al., 2015a*; *Kim et al., 2015b*; *Kim and Chen, 2019*; *Lee and Chen, 2019*). The current study has revealed a similar role for invasive protrusions in promoting myoblast fusion during adult skeletal muscle regeneration, demonstrating that the same cell fusion machinery required during embryogenesis is reused in adult muscle regeneration. It is striking that depleting the branched actin polymerization machinery results in a severe SCM fusion defect similar to depleting the fusogenic protein MymX, highlighting the indispensable role for actin cytoskeletal rearrangements in SCM fusion. Indeed, our previous work with a reconstituted cell-fusion culture system led to the discovery that fusogens and branched actin regulators are two minimal components of the cell–cell fusion machinery, and that actin-propelled invasive protrusions are required to bring the two apposing cell membranes into close proximity for fusogen engagement (*Shilagardi et al., 2013*). It would be interesting to determine whether invasive protrusions promote the trans-interactions of fusogens at the mammalian fusogenic synapse.

## Materials and methods
### Mouse genetics

C57BL/6J (stock: 000664) and *Pax7*[CreERT2] (stock: 012476) (*Lepper et al., 2009*) mice were obtained from the Jackson Laboratory. The *Arpc2*[fl/fl] (*Wang et al., 2016*), *Nwasp*[fl/fl] (*Cotta-de-Almeida et al., 2007*), and *Cyfip1*[fl/fl] (*Habela et al., 2020*) mice were previously described. The *Mymx*[fl/fl] line (*Bi et al., 2018*) was generously provided by Dr. Eric N. Olson. The control and mutant male littermates were used in each cohort of experiments.

### Tamoxifen and BaCl$_2$-induced muscle injury

Tamoxifen (Sigma; T5648) was dissolved at 20 mg/ml in corn oil. 100 μl tamoxifen/corn oil solution was administered by intraperitoneal injection to 2-month-old male mice as schematized in the figures. To induce muscle injury, BaCl$_2$ (Sigma; 342920) was dissolved in sterile saline to a final concentration of 1.2%, aliquoted, and stored at −20°C. Mice were anesthetized by isoflurane inhalation, the legs were shaved and cleaned with alcohol, and TA muscles were injected with 50 μl of BaCl$_2$ with a 28-gauge needle.

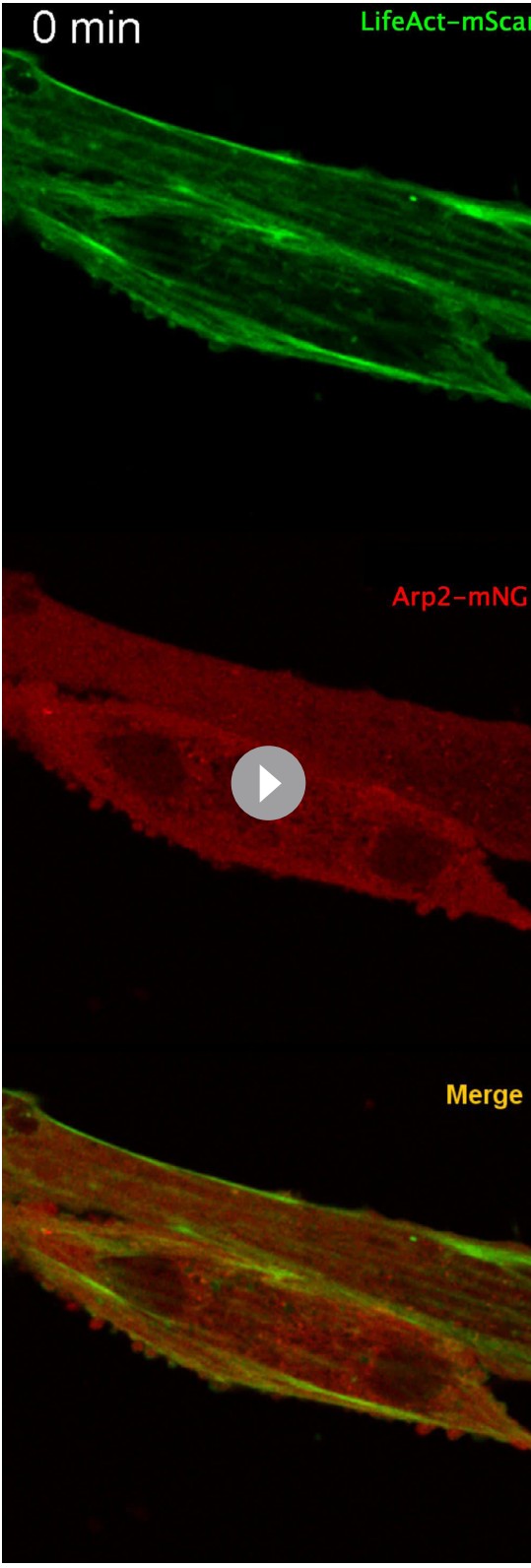

**Video 4.** F-actin and the Arp2/3 complex are enriched in the invasive protrusions at the fusogenic synapse of SCMs. Time-lapse imaging of a fusion event between two mouse SCMs co-expressing LifeAct-mScar and Arp2-mNG at 24 hours in DM. Note that F-actin and Arp2 were enriched in the finger-like invasive protrusions at the fusogenic synapse (arrows) and dissolved immediately after cell membrane fusion. The time interval is two minutes. Single focal plane is shown.

https://elifesciences.org/articles/103550/figures#video4

## Satellite cell isolation and culture

Satellite cells were isolated from limb skeletal muscles of 2-month-old male mice. Briefly, muscles were minced and digested in 800 U/ml of type II collagenase (Worthington; LS004196) in F-10 Ham's medium (Thermo Fisher Scientific; 11550043) containing 10% horse serum for 90 minutes at 37°C with rocking to dissociate muscle fibers and dissolve connective tissues. The dissociated myofiber fragments were collected by centrifugation and digested in 0.5 U/ml dispase (Gibco; 17105041) in F-10 Ham's medium for 30 minutes at 37°C with rocking. Digestion was stopped with F-10 Ham's medium containing 20% FBS. Cells were then filtered from debris, centrifuged, and resuspended in growth medium (GM: F-10 Ham's medium supplemented with 20% FBS, 4 ng/ml FGF2, 1% penicillin–streptomycin and 10 mM HEPES). The cell suspension from each animal was pre-plated twice on the regular 100 mm tissue culture-treated dishes for 30 minutes at 37°C to eliminate fibroblasts. The supernatant containing mostly myoblasts was then transferred into collagen-coated dishes for culture in GM. To validate the KO efficiencies of the target genes, skeletal muscle from one to six mice of each genotype was pooled for satellite cell isolation. To induce myogenic differentiation, satellite cells were cultured in DM (DMEM supplemented with 2% horse serum, 1% penicillin-streptomycin, and 10 mM HEPES).

## Pharmacological treatments of satellite cells

To pharmacologically inhibit branched actin polymerization in SCMs, the Arp2/3 inhibitor CK666 (50 μM) was added into the DM at day 0 of differentiation of wild-type SCMs. After 48 hours, the cells were fixed in 4% paraformaldehyde (PFA) and stained with anti-MHC and DAPI to assess their differentiation and fusion index.

To delete *Arpc2* in SCMs in vitro, satellite cells isolated from *Arpc2*-cKO mice were cultured in GM supplemented with 2 μM 4-hydroxytamoxifen (Sigma; H6278) for 10 days. Subsequently, the cells were trypsinized and plated at 70% confluency in DM. After 48 hours, the cells were fixed in 4% PFA and stained with anti-MHC and DAPI to assess their differentiation and fusion index.

## Retroviral vector preparations and expression

The cytosolic GFP, cytosolic mScarleti, LifeAct-mScarleti, and Arp2-mNeongreen constructs were described in the previous study (*Lu et al., 2024*), and assembled into the retroviral vector pMXs-Puro (Cell Biolabs; RTV-012) using the NEBuilder HiFi DNA Assembly Cloning Kit (NEB; E2621L). To package the retrovirus, 2 μg of retroviral plasmid DNA was transfected into platinum-E cells (Cell Biolabs; RV-101) using the FuGENE HD transfection reagent (Promega, E2311). Two days after transfection, the virus-containing medium was filtered and concentrated with Retro-X Concentrator (Clontech, PT5063-2) following the manufacturer's protocol. The concentrated retroviruses were diluted in GM (with a 1:1000 dilution), mixed with polybrene (7 μg/ml), and used to infect cells. One day after infection, cells were washed with PBS and cultured in fresh GM.

## Immunohistochemistry

To co-stain NCAM, MAC-2, and Laminin, the 4% PFA fixed TA muscles were dehydrated in 30% sucrose at 4°C overnight. The specimens were embedded in Tissue-Plus O.C.T. Compound (Fisher Scientific; 23-730-571) and 12 μm cryosections were collected onto Superfrost Plus Microscope Slides (Fisher Scientific; 12-550-15). Then, the cryosections were incubated with blocking buffer (PBS containing 2% BSA and 0.1% Triton X-100) for 20 minutes at room temperature (RT), followed by overnight incubation with rabbit anti-NCAM (1:200; Millipore; AB5032), rat anti-MAC-2 (1:200; Biolegend; 125401), and rat anti-Laminin-2 (1:500; Sigma; L0663) at 4°C. To stain for dystrophin, the freshly dissected TA muscles were snap frozen in Tissue-Plus O.C.T. Compound and 12 μm cryosections were collected onto Superfrost Plus Microscope Slides. Next, the sections were fixed in 4% PFA for 12 minutes at RT, washed three times with PBS, and incubated with blocking buffer for 20 minutes at RT, followed by overnight incubation with rabbit anti-dystrophin (1:200; Abcam; ab15277) at 4°C. To co-stain Pax7, MyoG, Laminin, and Ki67, the freshly dissected TA muscles were snap frozen in Tissue-Plus O.C.T. Compound and 12 μm cryosections were collected onto Superfrost Plus Microscope Slides. Then, the sections were fixed in 2% PFA for 5 minutes at RT, washed three times with PBS, and incubated with blocking buffer supplemented with M.O.M blocking reagent (1:25; Vector; MKB-2213-1) for 60 minutes at RT, followed by overnight incubation with mouse anti-Pax7 (1:2; DSHB; Pax7), mouse anti-MyoG (1:2; DSHB; F5D), rat anti-Laminin-2 (1:500; Sigma; L0663), and rat anti-Ki67 (1:500; Thermo

Fisher Scientific; 14-5698-82) at 4°C. After the incubation with primary antibodies, the sections were extensively washed with PBS and then incubated with Alexa Fluor-conjugated secondary antibodies for 1 hour at RT. Subsequently, the sections were washed with PBS and subjected to imaging using a Leica TCS SP8 inverted microscope.

## Western blot

For western blots, proteins were isolated from the cultured SCMs or TA muscle using ice-cold RIPA buffer (150 mM NaCl, 1% NP40, 0.1% SDS and 50 mM Tris, pH 7.4) containing protease and phosphatase inhibitors (Cell Signaling Technologies; 5872) for 20 minutes. The supernatants were collected by centrifugation at $140,000 \times g$ for 15 minutes. Protein concentrations were determined using the Bradford Protein Assay Kit (Bio-Rad; 5000201). 10–30 µg total protein was loaded for each sample and separated by 10% SDS-PAGE gel and transferred to PVDF membranes (Millipore; GVHP29325). Then, the membranes were blocked for 1 hour at RT in PBS containing 5% nonfat dry milk and 0.1% Tween-20 (PBSBT) and subsequently were incubated with primary antibodies diluted at 1:1000 in PBSBT overnight at 4°C. The membranes were then washed with PBST and incubated with appropriate HRP-conjugated secondary antibodies diluted in PBSBT for 1 hour at RT. After extensive washes with PBST, the membranes were developed with the ECL western blotting substrate (Thermo Fisher Scientific; 32209). The following primary antibodies were used: sheep anti-ESGP/MymX (1:1000; R&D Systems; AF4580), mouse anti-MymK (*Zhang et al., 2020b*) (1:1000), and rabbit anti-β-Tubulin (1:1000; Cell Signaling Technologies; 2146).

## Time-lapse imaging and analysis

Time-lapse imaging of cells incubated in 5% $CO_2$ at 37°C was performed on a Nikon A1R confocal microscope with a Nikon Biostation CT. The satellite cells were seeded on fibronectin-coated cover glass (MATTEK; P35G-0-14C) and imaged using a 40× (0.4 NA) objective at indicated time points after switching from GM to DM. The cells were imaged at 2- or 5-minute intervals. After time-lapse imaging, ImageJ (NIH, 64-bit Java 1.8.0_172) was used to project the z-stacks in 2D, using maximum intensity projection, and the resulting 2D images were assembled into a time-lapse video.

## Electron microscopy

To observe the invasive protrusions at the contact sites of SCMs during muscle regeneration in vivo, TA muscle at dpi 3.5 was fixed in a solution containing 3% PFA, 2% glutaraldehyde, 1% sucrose, 3 mM $CaCl_2$ in 0.1 M sodium cacodylate buffer (pH 7.4) overnight at 4°C. Samples were subsequently washed with 0.1 M cacodylate buffer containing 3% sucrose and 3 mM $CaCl_2$, and post-fixed with 1% osmium tetroxide in 0.1 M sodium cacodylate buffer for 1.5 hours on ice. The muscle samples were stained with 2% uranyl acetate, dehydrated, and embedded in EPON resin as previously described (*Zhang and Chen, 2008*). The embedded samples were then cut into 70-nm-thick sections using LEICA ultramicrotome (UC6) and collected on copper slot grids. These sections were post-stained with 2% uranyl acetate and Sato's lead solution and examined using a JEOL 1400 transmission electron microscope.

## Statistics and reproducibility

Statistical significance was determined using a two-tailed Student's $t$-test conducted using the GraphPad Prism 8 software. The sample sizes and number of replicates are indicated in the figure legends. All experiments were repeated in at least three independent biological replicates. The investigators were not blinded to allocation during the experiments and outcome assessment. No data were excluded from the analyses. For the in vivo studies, age-matched animals were randomly assigned to experimental and control groups. No statistical methods were used to predetermine sample sizes, but our sample sizes are similar to those reported in previous publications (*Millay et al., 2014*; *Bi et al., 2018*; *Duan et al., 2018*). Data distribution was assumed to be normal, but this was not formally tested.

## Acknowledgements

We thank Dr. Eric Olson for generously providing the Mymx[fl/fl] mice, UT Southwestern Animal Resource Center for assistance with mouse colony maintenance, and UT Southwestern Quantitative

Light Microscopy Core Facility for assistance with 3D reconstruction of confocal images. This work was supported by an NIH grant (R35GM136316) to EHC. YL was supported by an American Heart Association Career Development Award (25CDA1451113).

## Additional information

### Funding

| Funder | Grant reference number | Author |
|---|---|---|
| American Heart Association | 10.58275/aha. 25cda1451113.pc.gr.229681 | Yue Lu |
| National Institute of General Medical Sciences | R35GM136316 | Elizabeth H Chen |

The funders had no role in study design, data collection and interpretation, or the decision to submit the work for publication.

### Author contributions

Yue Lu, Conceptualization, Data curation, Formal analysis, Funding acquisition, Investigation, Methodology, Writing – original draft, Writing – review and editing; Tezin Walji, Pratima Pandey, Methodology; Chuanli Zhou, Data curation; Christa W Habela, Scott B Snapper, Rong Li, Resources; Elizabeth H Chen, Conceptualization, Supervision, Funding acquisition, Investigation, Writing – original draft, Writing – review and editing

### Author ORCIDs

Yue Lu ⓘ https://orcid.org/0000-0002-3999-7050
Chuanli Zhou ⓘ https://orcid.org/0000-0003-1848-2055
Rong Li ⓘ https://orcid.org/0000-0002-0540-6566
Elizabeth H Chen ⓘ https://orcid.org/0000-0003-2707-6083

### Ethics

All animal studies were approved by the UT Southwestern Medical Center Animal Care and Use Committee according to NIH guidelines.

Reviewer #1 (Public review): https://doi.org/10.7554/eLife.103550.4.sa1
Reviewer #2 (Public review): https://doi.org/10.7554/eLife.103550.4.sa2
Reviewer #3 (Public review): https://doi.org/10.7554/eLife.103550.4.sa3
Author response https://doi.org/10.7554/eLife.103550.4.sa4

## Additional files

### Supplementary files
MDAR checklist

### Data availability

The data supporting the findings of this study are available within the article and its supplementary files. The materials used in this study are available from the corresponding authors upon reasonable request.

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
