## [Editor Report · eLife Assessment]

This study presents a **valuable** finding regarding the role of Arp2/3 and the actin nucleators N-WASP and WAVE complexes in myoblast fusion. The data presented is **convincing**, and the work will be of interest to biologists studying skeletal muscle stem cell biology in the context of skeletal muscle regeneration.

---

## [Referee Report · Reviewer #1 (Public review)]

Overall, the manuscript reveals the role for actin polymerization to drive fusion of myoblasts during adult muscle regeneration. This pathway regulates fusion in many contexts, but whether it was conserved in adult muscle regeneration remained unknown. Robust genetic tools and histological analyses were used to convincingly support the claims.

---

## [Referee Report · Reviewer #2 (Public review)]

To fuse, differentiated muscle cells must rearrange their cytoskeleton and assemble actin-enriched cytoskeletal structures. These actin foci are proposed to generate mechanical forces necessary to drive close membrane apposition and the fusion pore formation. While the study of these actin-rich structures has been conducted mainly in drosophila and in vertebrate embryonic development, the present manuscript present clear evidence this mechanism is necessary for fusion of adult muscle stem cells in vivo, in mice. The data presented here clearly demonstrate that ARP2/3 and SCAR/WAVE complexes are required for differentiating satellite cells fusion into multinucleated myotubes, during skeletal muscle regeneration.

---

## [Referee Report · Reviewer #3 (Public review)]

This manuscript addresses an important biological question regarding the mechanisms of muscle cell fusion during regeneration. The primary strength of this work lies in the clean and convincing experiments, with the major conclusions being well-supported by the data provided.

The authors have satisfactorily addressed my inquiries.

---

## [Author Response]

The following is the authors’ response to the previous reviews.

**Reviewer #3 (Public review):**
The authors have satisfactorily addressed my inquiries. However, I had to look quite hard to find where they responded to my final comment regarding the potential role of Arpc2 post-fusion during myofiber growth and/or maintenance, which I eventually located on page 7. I would appreciate it if the authors could state this point more explicitly, perhaps by adding a sentence such as "However, we cannot rule out the possibility that Arpc2 may also play a role in....." to improve clarity of communication.While I understood from the original version that this issue falls beyond the immediate scope of the study, I believe it is important to adopt a more cautious and rigorous interpretative framework, especially given the widespread use of this experimental approach. In particular, when a gene could potentially have additional roles in myofibers, it may be helpful to explicitly acknowledge that possibility. Even if Arpc2 may not necessarily be one of them, such roles cannot be fully excluded without direct testing.

We appreciate the reviewer’s comments and have included several sentences at the end of the “Branched actin polymerization is required for SCM fusion” section to address this question:

“The severe myoblast fusion defects observed in early stages of regeneration (e.g. dpi 4.5) provide a good explanation for the presence of thin muscle fibers in ArpC2 cKO mice at dpi 14 (Fig. 2B and 2C) and dpi 28 (Fig. S4A and S4B). These thin muscle fibers could be either elongated mononucleated muscle cells or multinucleated myofibers each containing a small number of nuclei due to occasional fusion events (comparable to those in Myomixer cKO muscles) (Fig. 2B and 2C; Fig. S4A and S4B). Whether Arp2/3 and branched actin polymerization play a role in the growth and/or maintenance of post-fusion multinucleated myofibers requires future loss-of-function studies in which ArpC2 cKO is generated using a myofiber-specific cre driver.”